# Effect of motivated physicians and elderly patients with hypertension or type 2 diabetes mellitus in prepared communities on health behaviours and outcomes: A population-based PS matched retrospective cohort study during five-year follow-up period

Eun Jee Park[1]☯, Hyunsung Kim[2]☯, Yaeji Lim[2], Soon Young Lee[3]*, Weon-Young Lee[1]*

1 Department of Preventive Medicine, Chung-Ang University College of Medicine, Seoul, Republic of Korea, 2 Department of Applied Statistics, Chung-Ang University, Seoul, Republic of Korea, 3 Department of Preventive Medicine and Public Health, Ajou University School of Medicine, Suwon-si, Korea

☯ These authors contributed equally to this work.
* wylee@cau.ac.kr (WYL); solee5301@gmail.com (SYL)

**Data Availability Statement:** The database used in this study was provided by the Korean National

## Abstract

Effective chronic disease management requires the active participation of patients, communities, and physicians. The objective of this study was to estimate the effectiveness of the Community-based Registration and Management for elderly patients with Hypertension or Type 2 Diabetes mellitus Project (CRMHDP) by using motivated primary care physicians and patients supported by prepared communities, to utilise healthcare and health outcomes in four cities in South Korea. We conducted a propensity score-matched retrospective cohort study using 2010–2011 as the baseline years, alongside a follow-up period until 2015/2016, based on the Korean National Health Insurance database. Both a CRMHDP group (n = 46,865) and a control group (n = 93,730) were applied against healthcare utilisation and difference-in-differences estimations were performed. For the health outcome analysis, the intervention group (n = 27,242) and control group (n = 54,484) were analysed using the Kaplan–Meier method and Cox proportional hazard regression. Results: The difference-in-differences estimation of the average annual clinic visits per person and the average annual days covered were 1.26 (95% confidence interval, 1.13–1.39) and 22.97 (95% CI, 20.91–25.03), respectively, between the intervention and control groups. The adjusted hazard ratio for death in the intervention group, compared to the control group, was 0.90 (95% CI, 0.86–0.93). For stroke and chronic renal failure, the adjusted hazard ratios for the intervention group compared to the control group were 0.94 (95% CI, 0.88–0.99) and 0.80 (95% CI 0.73–0.89), respectively. Our study suggests that for effective chronic disease management both elderly patients and physicians need to be motivated by community support.

Health Insurance Service (NHIS), and it is stipulated not to be distributed to the public by law. Therefore, it is third-party data that the authors cannot legally distribute. To be more specific in terms of the data acquisition process, researchers who aim to use the NHIS data for their research must access the NHIS's data sharing service webpage (https://nhiss.nhis.or.kr), obtain the approval for use, and pay the data usage fees. The research management number approved in this study is NHIS-2020-1-178. It is regulated by the NHIS that the approved data be used only for research purposes, and public sharing and use for other purposes be strictly prohibited. In the event of a problem in violation of this regulation, it is stipulated to take civil and criminal liability.

**Funding:** This study was conducted as part of hypertension and type 2 diabetes registration and management effectiveness evaluation analysis study in Gyeonggi-do 4 regions (R-2019-14) with the support of the Gyeonggi Public Health Policy Institute. The funding bodies had no role in the design of the study, data management, analysis, manuscript preparation, or decision to submit the results.

**Competing interests:** The authors have declared that no competing interests exist.

## Introduction

Chronic diseases often require a long period of supervision, observation, or care. The defining features of primary care, such as continuity, coordination, and comprehensiveness, make it suitable for managing chronic conditions [1]. To improve the quality of healthcare being delivered for chronic diseases in a primary care setting, payers have frequently used the pay-for-performance plan, which rewards physicians and medical groups that meet specific performance targets, while also offering financial incentives. In the United Kingdom, a Quality and Outcomes Framework (QOF) was introduced for all primary care physicians in 2004 [2], while in the United States, almost all pay-for-performance programs include incentives for primary care physicians [3]. However, studies evaluating such schemes have reported either a small positive effect or no effect on patient outcomes [4–7]. Thus, it may need to be considered that effective chronic disease management requires the active participation of patients and communities in addition to healthcare providers.

According to the chronic care model, high-quality chronic care is characterised by productive interactions between physicians and patients, which are more likely to be productive if the patients are active [4, 8–10]. In primary care practices, shared financial incentives for physicians and patients resulted in significantly reducing LDL-C levels at 12 months; however, incentives awarded to physicians or patients alone did not [11]. Moreover, to improve chronic care, healthcare provider organisations require associations with community-based resources through exercise programs and senior centres [4, 12]. Thus, community linkages would be especially helpful for small physician offices with limited resources.

The Korean Ministry of Welfare and Health (KMWH) and the Korean Disease Control and Prevention Agency (KDCA) established a pilot project in 2009, which was a Community-based Registration and Management of Hypertension and Type 2 Diabetes mellitus Project (CRMHDP) that activated both primary care physicians and patients in Gwangmyeong City [13]. Subsequently, this project has been expanded to include 21 urban and rural areas throughout the nation and has been viewed as an effective and appropriate chronic disease care model, in Korean primary care settings, by health policy professionals affiliated with the Organization for Economic Cooperation and Development (OECD) [14]. This study aimed to examine the impact of the CRMHDP on healthcare use and the health outcomes of patients using a population-based propensity score-matched retrospective cohort study over eight years.

## Materials and methods

### Context and interventions

Everyone in Korea is eligible for coverage under the National Health Insurance Service (NHIS), as a single insurer. Every hospital and local clinic should provide them with healthcare services, which are covered by the NHIP and are reimbursed based on a fee-for-service schedule. The insured individual makes contributions to the National Health Insurance Corporation (NHIC), which manages NHIS and is also required to pay a certain portion of the healthcare costs when they use any healthcare services. A patient with a chronic disease pays co-payments for a consultation with the physician at a local clinic and the drugs, covered by the NHIS, from a pharmacy. Primary care systems in Korea have some disadvantages in chronic disease management, whereby primary care physicians do not have enough financial incentives to provide primary care services to patients with chronic diseases, following national clinical guidelines for hypertension and type 2 DM. Moreover, patients with chronic diseases are likely to go 'doctor shopping' and to an outpatient clinic at hospitals, ultimately, passing by the local clinics

[14]. Lastly, most local primary care clinics do solo practices and have only one to two nurses. Every city and county in Korea has a public health centre (PHC), which is operated by the local government and whose chief role is to implement health promotion programs, including group education for the residents in the local government boundary. However, there are no collaborations and relationships between local clinics and the PHC for chronic disease management that are based in the community in Korea.

CRMHDP (S1 Fig) included some incentives to encourage the active participation of both elderly patients and physicians in this project. Elderly patients with hypertension or type 2 DM were registered to the CRMHDP by their regular local private clinics. They received a financial incentive, whereby they were exempt from the monthly co-payment (USD 1.50) for a clinic visit and from the co-shared cost (USD 2.00) per prescription for one month, for the drugs, which are insured by the NHIS. As a non-financial incentive for the enrolees, appointment reminder services for each scheduled clinic visit were provided to each registered patient. If they did not visit the doctor for three months, the Registration and Education Centre (REC) at the PHC, which had five staff members (a team leader, two nurses, and two dieticians), made phone calls to enquire about any potential problems. Participating physicians were given a small monetary incentive (USD 1) for registering a patient to the CRMHDP via their clinics, while the registration helped to maintain their current patient list and to attract new patients in the Korean primary care setting, with a low gatekeeping function. The non-financial incentive for physicians was the minimisation of their additional workload, which would have increased from being involved in the project engagement. Here, participating physicians entered only the basic health information required into the registration forms when enrolees visited their clinics. Moreover, if a registered patient was required to receive intensive education sessions on diet, exercise, and diseases due to poor control of HbA1c, a personalised counselling session was provided to the referred patient by the REC team in the PHC instead of the primary physicians. A REC at each PHC hosted two mass education sessions for the enrolees on self-management, which lasted one hour per session. Moreover, it participated in a mass campaign to make local residents recognise the importance of early detection and management of chronic diseases.

## Ethics approval

Ethical approval for this project was received from the human ethics committees of Chung-Ang University (1041078-201910-ZZSB-314-01), according to the guidelines of the Declaration of Helsinki. Following the ethics of the committees, individual consent was no longer required for the purpose of data linkage and evaluation.

## Study design and sample

To evaluate the CRMHDP, we conducted a retrospective cohort study using a propensity score (PS) matched control group design, whereby 2010–2011 was used as the baseline years, while the follow-up period was until 2015–2016. The study was based on routine health insurance data claimed through the NHIS database. The NHIS represents a mandatory, single, social health insurance, which covers the whole population of Korea. Patient information was anonymised and de-identified prior to analysis. Fig 1 shows a flowchart of study subjects. Inclusion criteria for study subjects included (1) residence in Gyeonggi-do, (2) age restrictions (65 and older), and (3) experience, where they have visited local clinics in Gyeonggi-do at least twice to receive treatment for hypertension (ICD-10 code: I10) or type 2 DM (ICD-10: E2) during the baseline years. Exclusion criteria in this study included (1) being diagnosed with a catastrophic, rare, and intractable disease as a comorbidity (i.e., cancer, stroke, or myocardial

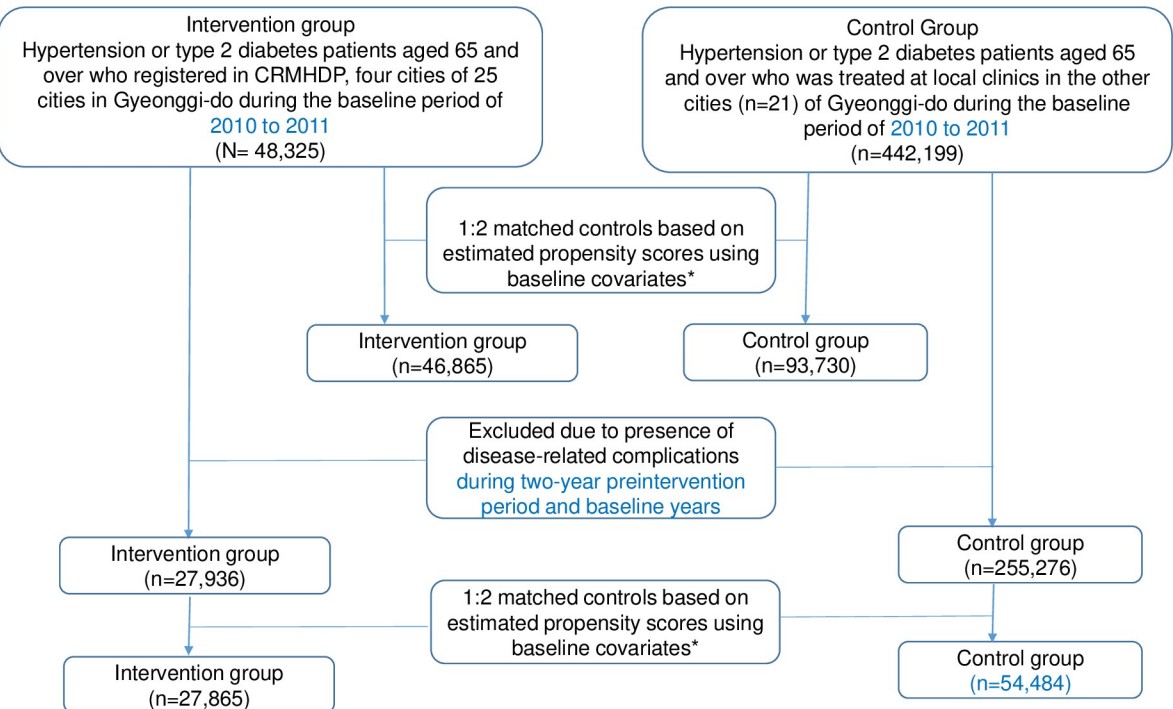

**Fig 1. Flowchart of study subjects.** CRMHDP: Community based Registration and Management for Hypertension and type 2 Diabetes Mellitus Project * Covariates: gender, age, income level, presence of coexisting condition, type of physician speciality, type of public health security and medication adherence during the past two years before baseline years.

infarction) during the baseline years, (2) died persons during the baseline years or moving out of Gyeonggi-do during the follow-up period. In the NHIS database, the identification of patients with hypertension, type 2 DM, or rare and intractable diseases was based on the presence of the ICD-10 code, such as I10 [15], E2 (type 2 DM), or V (rare and intractable diseases) in the diagnosis form.

Four cities (Gwangmyeong-si, Hanam-si, Namyangju-si, and Ansan-si) in the Gyeonggi provincial area (Gyeonggi-do) participated in the project (Fig 1). To evaluate the impact of the intervention on annual local clinic visits and medication adherence, the CRMHDP group (n = 48,325), designated as the intervention group, was composed of patients who enrolled in the project via participating local clinics (n = 409)which accounted for 84.2% of all local clinics (n = 486), in the cities, during the baseline period (September 1, 2010, to December 31, 2011). The local clinics not engaged in the project had few patients with hypertension and type 2 DM because their clinical practioners were specialist such as opthamologist, pediatric physician, otolaryngologist and opthalmologis. Therefore, the registered patients can represent all elderly hypertensive and diabetic patient in the cities. The control group (n = 442,199) comprised patients who were treated at all local clinics (n = 2,860), in the other cities (n = 21) of Gyeonggi-do, which did not participate in the project. To assess the impact on the hospital admissions by major complications from hypertension or type 2 DM, participants with complications such as angina (I20), myocardial infarction (I21–I23), stroke (I60–I69), chronic renal failure (N17–N19), a glomerular disorder in diabetes (N09.3), vascular disease (I70–I79), hypertensive retinal disease (H35.02), diabetic retinal diseases (H36.02), diabetic polyneuropathy (G63.2), and a diabetic ulcer (E1470) during the two-year pre-intervention period and baseline years were excluded from both the CRMHDP group and the control group. Thus, the

sample sizes of the CRMHDP and the control groups were 27,736 and 255,276, respectively, to evaluate the impact on hospitalisations owing to disease-specific complications.

## Study variables

**Outcome variables.**   Primary outcomes for the medical effectiveness of the CRMHDP were continuous treatment and medication adherence, which were used to assess the short-term effect of the project against mortality and hospital admissions due to major complications of hypertension and type 2 DM for the evaluation of the long-term effect. The indicator for continuous treatment was the annual visits to the local clinic per person, for treatment of hypertension or type 2 DM, while the medication adherence was measured by the proportion of days covered (PDC) for each antihypertensive drug or antidiabetic drug or insulin injection during each period. This approach provided a true picture of the days on which a patient was covered with medication, rather than a simple summation of the days' supply for all fills divided by the number of days in a particular period, defined as the medication possession ratio (MPR), which accounted for overlapping refills [16]. For the long-term outcomes, we employed an all-cause mortality and hospital admission (including admission from emergent service) due to major complications in hypertension or type 2 DM, such as myocardial infarction (I20–I25), stroke (I60–65), and chronic renal failure (N17–N19).

**Independent variables.**   The sociodemographic and clinical characteristics of the study's participants were compared between the intervention group and the control group. The sociodemographic variables were considered as gender, age, household income level, and type of social health security. Household income levels were calculated using five equal brackets of household income and measured by income-related contributions from the Korean NHIS. Social health security was composed of the National Health Insurance Service and subscribed for by the employed–insured and the self-employed and medical aid program to provide low-income groups with healthcare services, using government subsidies. The clinical variables were the presence of the coexistence of hypertension or type 2 DM, the past medication adherence prior to participation in the study, and the type of speciality by the physician treating the patients in the study. The past medication adherence rates during the two-year per-intervention period, were measured by the PDC. The specialities of the physicians were classified into either an internal medicine and family medicine group or another group. The trends for the annual visits to the clinic and the number of annual days covered for antihypertensive or antidiabetic drugs per person were compared between the intervention and control groups during the study follow-up period (2010/11–2015/16).

## Statistical analysis

The analysis was based on propensity score matching(PSM) which was first proposed by Rosenbaum and Rubin in the 1980s [17]. PSM is to make the selected two groups comparable in terms of potential confounding factors, in order to balance variables and reduce bias. The PS, defined as the probability of participation in the CRMHDP conditional on baseline covariates, was calculated using a multivariate logistic regression using CRMHDP participation as the dependent variable. For the independent baseline covariates, the patient's sociodemographic characteristics (sex, age, type of public health insurance, and household income), the patient's clinical characteristics (presence of hypertension or/and type 2 DM, compliance of antihypertensive and antidiabetic drugs type 2 DM during the two years prior to baseline years), and healthcare provider characteristics (type of physician speciality) were included to reflect their health status at the baseline. Based on the estimated PS, two controls were matched to every CRMHDP participant to increase precision by using a near-neighbour-matching

algorithm without replacement, which was adapted from the SAS macro from Coca–Perraillon [18]. If there is missing in the variable or there is no control patient within the proper range of the propensity score, these data are eliminated all. Those data are not included in the neither intervention nor control group in this study. Therefore, there is no missing in the matching variables of the data used in the analysis. The PS calculation and the matching were performed and stratified by the baseline years 2010 and 2011. To assess the quality of the matching, i.e., if the covariates were balanced between the matched groups, the calculation of the standardised mean differences (SMDs) between the groups was performed before and after matching. A SMD close to zero indicates a good balance of the covariate between the CRMHDP group and the control, while 0.1 was recommended as the threshold for declaring imbalance [19].

As shown in Fig 1, among the study participants for the impact assessment of the intervention on local clinic visits and medication adherence, the intervention (n = 46,865) and control groups (n = 93,730) were extracted after matching. The SMDs of all covariates were less than 0.1, which indicates a good balance of covariates between them (S1 Table). To assess the impact of the intervention on hospital admissions involving major complications of hypertension and type 2 DM, the intervention (n = 27,242) and control groups (n = 54,484) were generated after matching and the SMDs of all covariates were less than 0.1 (S2 Table), thereby indicating a good balance of covariates between them (S2 Fig).

The demographics (gender, age, income status, and type of public health insurance) and clinical characteristics (disease coexistence, type of physician speciality, and medication adherence one and two years before the baseline period) were compared between the intervention and control groups. The annual visits to the clinic and the annual days covered by the antihypertensive drug, antidiabetic drug, or insulin injection per person were calculated during a period of two years prior to and five years after initial registration. To examine the effect of CRMHDP on the healthcare utilisation of primary care and medication adherence, the difference-in-differences (DiD) estimators, which is defined as the difference in the average outcome in the intervention group before and after intervention minus the difference in the average outcome in the control group before and after the intervention, were calculated based on linear regression analysis [20]. Moreover, DiD regression was implemented to test the interaction term between the intervention group and time, while adjusting for any covariates, such as gender, age, income status, type of public health insurance, disease coexistence, type of physician specialty, and medication adherence one and two years before the baseline period.

Differences in the all-cause mortality and hospital admissions due to complications from hypertension or type 2 DM between the intervention and control groups were examined using Kaplan–Meier method and Cox proportional hazard regression with adjustments for gender, age, income status, type of public health insurance, disease coexistence, type of physician, and medication adherence one and two years before the baseline period.

## Results

Analysis of the impact of the CRMHDP on local clinic visits and medication adherence included 46,865 CRMHDP group participants in the intervention group, while the control group sample, which was matched by a 1:2 ratio with the PS, included 93,730 participants. Study characteristics (gender, age, income status, disease coexistence, type of physician speciality, type of public health insurance, and medication adherence one and two years before the baseline period) were not significantly different between the groups (S3 Table). To assess the impact of the project on hospital admission due to major complications in hypertension or type 2 DM, the number of project group members was 27,242 and the control sample size was 54,484, using the same matching technique as before (S4 Table).

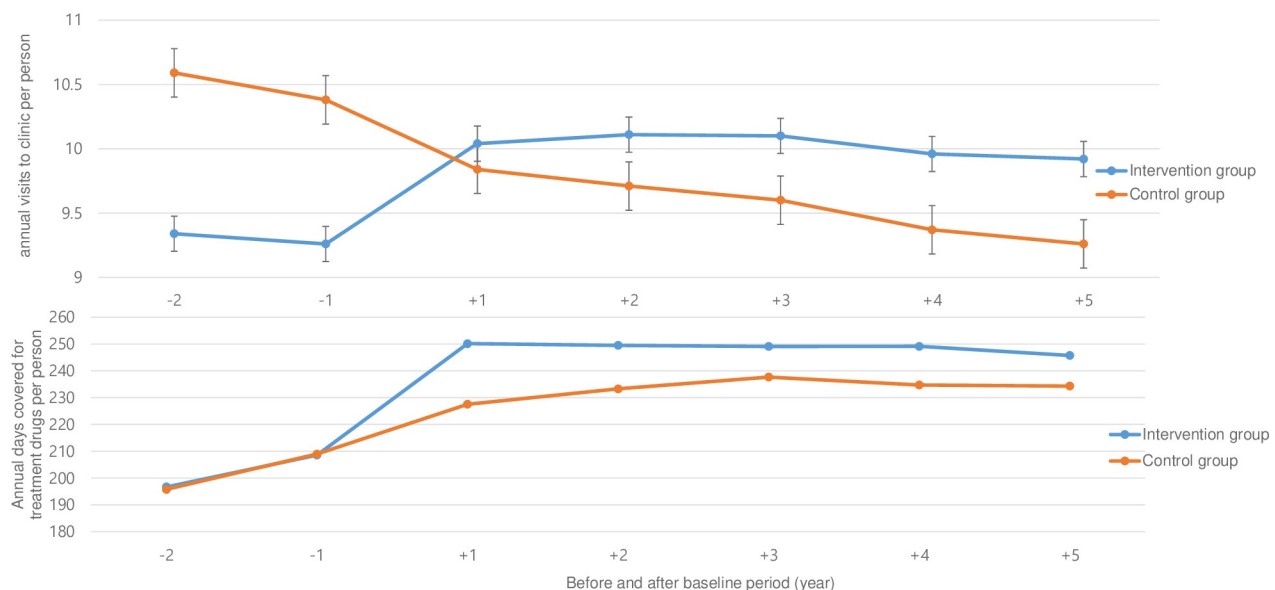

**Fig 2. Trends of annual visits to the clinics per person and annual days covered for antihypertensive, antidiabetic drugs, or insulin injections per person in intervention and control groups during the period of two years prior to and five years after the baseline period.**

The annual visits to local clinics per person in the intervention group surged one year before and after participation in the project and maintained at around 10 times thereafter, while those in the control group gradually decreased overtime (Fig 2). The annual days covered for either the antihypertensive drug, antidiabetic drug, or insulin injection per person in the intervention group increased prior to and post-project participation and further increased thereafter. Conversely, those in the control group steadily increased overtime, yet remained far behind the intervention group by the end of the follow-up period (Fig 2). While the average number of annual clinic visits per person increased by 0.78 in the intervention group one year before and after the participation in the project, it decreased by 0.48 in the control group over the same period.

The difference-in-differences estimate of the average annual clinic visits per person was 1.26 (95% confidence interval, 1.13–1.39) between the intervention and the control groups (Table 1). In the DiD regression estimate for the annual clinic visits per person, the interaction term between time (preintervention vs postintervention period) and group (intervention vs control group) demonstrated significant positive coefficients after adjusting for the confounding variables (S5 Table). This implied that the intervention contributed to escalate the annual clinic visits per person one year before and a year after the intervention. In the intervention group, the average number of days covered for the drugs increased by 41.56 in the years before and after the intervention and increased by 18.58 again over a similar period. The difference-in-differences estimate for the average annual days covered was 22.97 (95% CI, 20.91–25.03) between the intervention and the control groups (Table 1). The results of the DiD regression estimate for the days covered showed that the interaction term between the time (preintervention vs postintervention period) and the groups (intervention vs control group) had a significant positive coefficient after adjusting for any confounding variables (S5 Table). This meant that the intervention led to strengthen compliance in the annual days covered for antihypertensive, antidiabetic drug and insulin injections.

Fig 3 shows the survival curves within five years of the all-cause death and hospital admissions due to stroke, acute myocardial infarction, and chronic renal failure using the Kaplan–

**Table 1. Differences-in-differences estimates for the effect of CRMHDP on annual clinic visits per person and annual days covered for antihypertensive drugs, anti-diabetic drugs, and insulin injections per person in the control group matched by propensity scores.**

| Dependent Variables | Group | One year before and after participation | | |
|---|---|---|---|---|
| | | Preintervention period | Postintervention period | Postintervention–preintervention period differences |
| Average number of annual clinic visits per person | Intervention group (N = 46,865) | 9.26 (0.040) | 10.04 (0.037) | 0.78 (0.051) |
| | Control group (N = 93,730) | 10.32 (0.024) | 9.84 (0.024) | -0.48 (0.036) |
| | Intervention group and control group differences | -1.06 (0.047) | 0.20 (0.044) | 1.26 (0.064) |
| | T value | -23.17 *** | 4.83 *** | 19.64 *** |
| Annual days covered for drugs per person | Intervention group (N = 46,865) | 208.6 (0.597) | 250.1 (0.609) | 41.56 (0.814) |
| | Control group (N = 93,730) | 208.9 (0.422) | 227.5 (0.443) | 18.58 (0.625) |
| | Intervention group and control group differences | -0.37 (0.760) | 22.60 (0.719) | 22.97 (1.049) |
| | T value | -0.50 | 32.11 *** | 21.89 *** |

***$p < 0.01$, **$p < 0.05$, *$p < 0.1$.

Times, days (standard error)

Meier method. For all-cause mortality, while the survival probability of the intervention group in the past five years was 85.66% (95% CI, 85.24–86.07), for the control group over the same period, it was 84.13% (95% CI, 83.82–84.44%). The survival probabilities of the intervention and control groups for stroke in the past five years were 93.11% (95% CI, 92.76–93.51) and 92.51% (95% CI, 92.23–92.75), respectively. Similarly, for acute myocardial infarction, the survival probabilities for the intervention and control groups during the 5-year follow-up period were 94.05% (95% CI, 93.72–94.34) and 93.91% (95% CI, 93.59–94.21), respectively. The

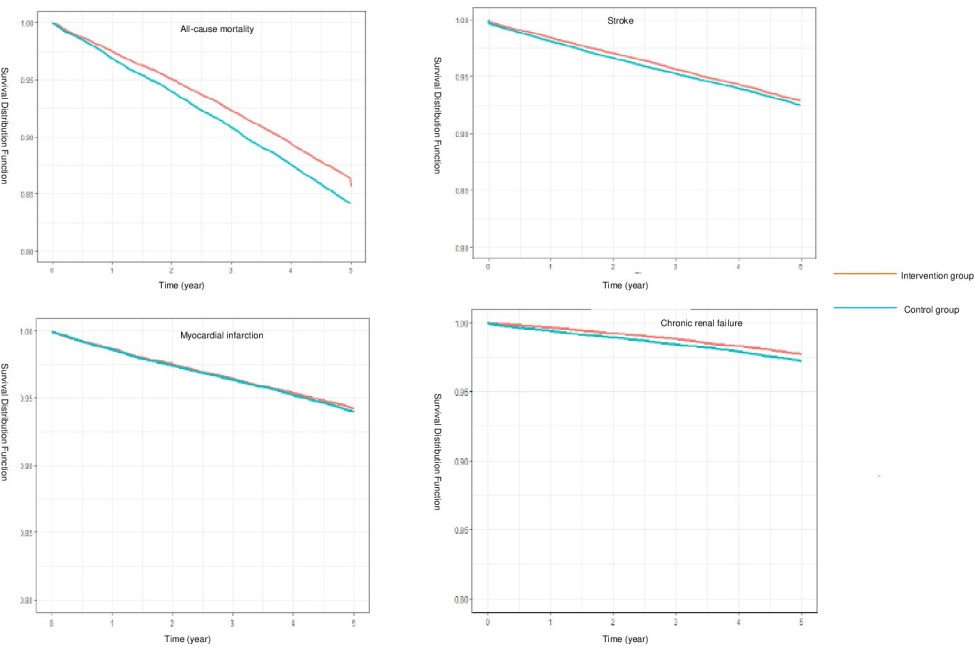

**Fig 3. Differences in all-cause mortality and complications (stroke, acute myocardial infarction, and chronic renal failure) related to hospitalisations between the intervention and control groups, using the Kaplan–Meier curve, during the 5-year follow-up period.**

**Table 2. Hazard ratios for the intervention group vs. the control group for all-cause death and complications (stroke, acute myocardial infarction, and chronic renal failure) during the five-year follow-up period based on the Cox proportional hazard model (among excluded the pre-existing complications).**

| | [a]Hazard ratios# | 95% Confidence Limits | | p |
|---|---|---|---|---|
| **All-cause death** | 0.897 | 0.864 | 0.932 | < .0001*** |
| **Stroke** | 0.935 | 0.884 | 0.989 | 0.0183* |
| **Myocardial infarction** | 0.962 | 0.904 | 1.024 | 0.2275 |
| **Chronic renal failure** | 0.803 | 0.728 | 0.885 | < .0001*** |

***$p < 0.001$, **$p < 0.05$, *$p < 0.$

# Adjusted variables: gender, age, income status, type of public health insurance, disease coexistence, type of physician, and medication adherence one and two years before the baseline period.

survival probabilities in the intervention and control groups during the last five years for chronic renal failure were 97.61% (95% CI, 97. 31–97.89) and 96.81% (95% CI, 96.52–97.19), respectively. The adjusted hazard ratio for death in the intervention group compared to the control group was 0.90 (95% CI, 0.86–0.93) (Table 2). For stroke and chronic renal failure, the adjusted HRs for the intervention group compared to the control group were 0.94 (95% CI, 0.88–0.99) and 0.80 (95% CI 0.73–0.89), respectively, while the adjusted HR for cardiovascular disease was not significant (Table 2).

## Discussion

This study showed that the implementation of the CRMHDP led to more local clinic visits and improved medication adherence, which could prevent all-cause mortality and hospital admission owing to complications from hypertension or type 2 DM, such as stroke and chronic renal failure except for acute myocardial infarction. Given that the primary care physician can offer chronic disease management and preventive care, more local visits occurred in the intervention group, which meant that the CRMHDP enrolees had more chances to screen blood pressure, perform HbA1c tests, and beneficial counselling from local clinics, to potentially improve health outcomes [21, 22]. Moreover, better medication adherence was associated with a lower occurrence of disease-specific complications of hypertension and type 2 DM [23–25] However, there was no significant difference in acute myocardial infarction-related hospitalisations between them. A potential reason for no significant differences being observed in our study is that the 5-year follow-up period might be not enough to observe the occurrence of complications associated with hypertension or type 2 DM.

There might be three reasons that the CRMHDP group had more clinical visits and improved medication adherence than the control group. Firstly, the introduction of reduced cost-sharing for healthcare use and medications could motivate patients with chronic diseases to improve their medication compliance [26]. Given that this financial support was likely to be more effective in low-income groups and that most elderly Koreans have economic difficulty due to the lack of public pension, a small exemption from out-of-pocket payments for routine clinic visits and medication in Korean primary care could help create desirable health behaviours in elderly patients with chronic diseases [27, 28]. Secondly, in spite of the local physicians' heavy workload from the numerous visits of patients to the clinics, almost all of them performed an active role in the project since the project did not place an extra burden on them, while the registration in the project and financial support ensured that their patients remained in their own clinics, meaning that a patient was registered with a primary care physician in a Korean primary care setting, where previously doctor shopping had prevailed. Thirdly, the Registration and Education Center (REC) at the Public Health Center (PHC)

provided recall and reminder services, individual counselling, and group education classes on blood pressure and glucose control, as a further community resource. Furthermore, this scheme might help busy local clinics relieve their workload. In this project, any enrolled patients that missed their clinic visits for more than two months received calls and reminder messages from the REC at the PHC. The recall and reminder messages could improve medication adherence for patients who require long-term medications [29].

This study has some limitations. First, where potential biases are likely to be greater for nonrandomised studies, such as this study, which used a retrospective cohort design compared to randomised trials in the evaluation of the effects of interventions; therefore, the results should always be interpreted with caution [30]. In general, a retrospective cohort study design is likely to provide poor control over any confounder [31]. For example, confounders, such as pre-existed co-morbidities besides hypertension and type 2 DM like COPD, hepatic and renal diseases were not included in the data analysis. However, patients diagnosed with catastrophic disease such as cancer, stroke, and myocardial infarction, rare and intractable disease were excluded to avoid confounding bias due to the differences in the presence of catastrophic disease between intervention and control group. Second, the 5-year follow-up period in this study might not be long enough to observe an endpoint of death or disease-specific complications associated with hypertension and type 2 DM. Given that the survival curve of the all-cause mortality and hospital admissions owing to disease-specific complications showed increasing disparities between the intervention and control groups overtime, an extension to the follow-up period can more clearly examine the impact of the CRMHDP on any disease-specific complications. Moreover, many studies that have observed an event of complications from hypertension or type 2 DM have used a follow-up period of more than five years [32–35]. Third, this study did not include emergency departments visits and hospital stays as another health outcome. Fourth, we did not examine the effect of the CRMHDP on hypertension and type 2 DM separately due to the following reasons. According to the huge existing literature [36–38] related to health behavioral economics, financial incentives to chronic patient and physician for the change of patient behavior has been very effective tool at changing behaviors in chronic condition related to medication adherence. Hence, this study assumed the CRMHDP would have positive impact on patient behavior including medication adherence among both hypertensive patients and diabetic patients. However, since our assumption of the CRMHDP's impact on them may be wrong, a sub-group analysis should be made in the further. Lastly, we did not consider the duration of the disease because it could not be identified in the KNHIS (Korean National Health Insurance) data. However, this study used the past medication rates in two-year pre-intervention period as a confounding factor examing the effect of CRMHDP on the process and health outcome of health care service, which means that this study considers before and after the implementation period for CRMHDP.

This study suggests that both patients and physicians should be motivated toward chronic disease management alongside community support for any chronic disease management to be effective and sustainable in the community, including in primary care settings.

## Supporting information

**S1 Fig. Outlines of Community based Registration and Management of Hypertension and type 2 Diabetes mellitus Project (CRMHDP).**
(TIF)

**S2 Fig. Box-plots for the quality of matching.**
(TIF)

**S1 Table. Balance in measured baseline variables before and after matching (included prior to any complications existing).**
(PDF)

**S2 Table. Balance in the measured baseline variables before and after matching (excluded prior to any complications existing).**
(PDF)

**S3 Table. Baseline patient demographic and clinical characteristics of propensity score-matched intervention and control patients in the study (included prior to any complications existing).**
(PDF)

**S4 Table. Baseline patient demographic and clinical characteristics of propensity score-matched intervention and control patients in the study (excluded prior to any complications existing).**
(PDF)

**S5 Table. Difference-in-differences regression, including covariates of effects of the CRMHDP on the annual visits to clinics and annual days covered per person, with the control group matched by propensity scores.**
(PDF)

## Acknowledgments

This study used NHIS-NSC data (NHIS-2020-1-178) from the National Health Insurance Service (NHIS).

## Author Contributions

**Conceptualization:** Soon Young Lee.

**Data curation:** Eun Jee Park, Hyunsung Kim.

**Formal analysis:** Eun Jee Park, Hyunsung Kim, Soon Young Lee.

**Funding acquisition:** Soon Young Lee.

**Project administration:** Weon-Young Lee.

**Supervision:** Yaeji Lim, Soon Young Lee, Weon-Young Lee.

**Validation:** Yaeji Lim.

**Writing – original draft:** Eun Jee Park.

**Writing – review & editing:** Soon Young Lee, Weon-Young Lee.

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
