## [Decision Letter · Decision Letter 0]

6 Jun 2023

PONE-D-23-13912

Effect of motivated physicians and elderly patients with hypertension or type 2 diabetes mellitus in prepared communities on health behaviours and outcomes: a population-based propensity score-matched retrospective cohort study over eight years

PLOS ONE

Dear Dr. Lee,

Thank you for submitting your manuscript to PLOS ONE. After careful consideration, we feel that it has merit but does not fully meet PLOS ONE’s publication criteria as it currently stands. Therefore, we invite you to submit a revised version of the manuscript that addresses the points raised during the review process.

We look forward to receiving your revised manuscript.

Kind regards,

Yee Gary Ang, MBBS MPH

Academic Editor

PLOS ONE

Journal Requirements:

Additional Editor Comments:

We have invited 2 reviewers who have made some comments and I invite you to address these comments before we proceed further.

Reviewers' comments:

Reviewer's Responses to Questions

**Comments to the Author**

1. Is the manuscript technically sound, and do the data support the conclusions?

Reviewer #1: Yes

Reviewer #2: Yes

2. Has the statistical analysis been performed appropriately and rigorously? 

Reviewer #1: Yes

Reviewer #2: Yes

3. Have the authors made all data underlying the findings in their manuscript fully available?

Reviewer #1: Yes

Reviewer #2: No

4. Is the manuscript presented in an intelligible fashion and written in standard English?

Reviewer #1: Yes

Reviewer #2: Yes

5. Review Comments to the Author

Reviewer #1: This is a retrospective cohort using propensity score-paired matched control study of a Community based Registration and Management of Hypertension and Type 2 Diabetes mellitus Project (CRMHDP) that applying financial incentives activating both primary care physicians/patients and public health promotion team resulted in a statistically significant difference for type 2 DM patients for short and long term health outcomes in certain cities of Korea.

Among the countries applying system of universal coverage healthcare, I appreciated the hard work and determination of experimentally delivering such a high quality of healthcare for the personalized control of type 2 DM in Korea. It is well designed in various domains of healthcare system including incentives, followup strategy and additional community support for the health promotion. This an excellent longterm healthcare example of type 2 DM for other regions and other countries as well.

There are few minor concerns to be mentioned:

1) The control group defined as the patient’s clinical characteristics (presence of hypertension or/and type 2 DM). It's better to exclude the patients with hypertension to have pure control group of type 2 DM.

2) The parameters of long term outcome could consider either the frequency of emergent service of the target patients experienced or their annual/total medical expenditures during the followup years to disclose the severity and longterm financial burden of disease.

3) The parameters using in the proipensity score-matched pair method could consider other pre-existed medical co-morbidities besides hypertension and type 2 DM, such as COPD, hepatic and renal diseases, etc.

4) The format of tables and figures needs to be revised as journal requested.

Reviewer #2: In this manuscript, Park et al. present findings indicating that patients with effectively managed hypertension and/or DM have better outcomes. Overall, the paper demonstrates good writing quality and grammatical accuracy. The results are clear and straightforward. However, there are concerns regarding the reported results that need to be addressed before considering the manuscript for publication.

Results:

1. Figure 1: After excluding patients with disease-related complications and applying 1:2 matching, the numbers in the intervention group and control group are 27865 and 93730, respectively. Should these numbers be 27242 and 54484, respectively?

2. Table 2: Hazard ratios were calculated based on the intervention group (n = 46865) versus the control group (n = 93730). It is important to note that myocardial infarction and stroke may have a tendency for recurrence in certain cohorts. Could you provide hazard ratios based on the intervention group (n = 27242) versus the control group (n = 54484), which excludes the pre-existing complications? Additionally, it would be valuable to include data on the outcomes including admission, emergency department visits, or hospital stays during the follow-up period when comparing these two groups.

6. PLOS authors have the option to publish the peer review history of their article (what does this mean?). If published, this will include your full peer review and any attached files.

Reviewer #1: No

Reviewer #2: No

---

## [Author Response · Author response to Decision Letter 0]

22 Jun 2023

Please find point-by-point responses to the reviewers’ comments and an amended version of the manuscript with changes highlighted in blue in the attached file on 'Response to reviewers'. We are very grateful to the reviewers for their positive and useful suggestions, which have significantly improved the paper. 

We hope you will find the paper acceptable for your journal.

---

## [Decision Letter · Decision Letter 1]

15 Aug 2023

PONE-D-23-13912R1Effect of motivated physicians and elderly patients with hypertension or type 2 diabetes mellitus in prepared communities on health behaviours and outcomes: a population-based propensity score-matched retrospective cohort study over eight yearsPLOS ONE

Dear Dr. Lee,

Thank you for submitting your manuscript to PLOS ONE. After careful consideration, we feel that it has merit but does not fully meet PLOS ONE’s publication criteria as it currently stands. Therefore, we invite you to submit a revised version of the manuscript that addresses the points raised during the review process.

Sorry for the delay but the original 2 reviewers were not available for the revision so we looked for 2 additional reviewers.==============================

We look forward to receiving your revised manuscript.

Kind regards,

Yee Gary Ang, MBBS MPH

Academic Editor

PLOS ONE

Reviewers' comments:

Reviewer's Responses to Questions

**Comments to the Author**

1. If the authors have adequately addressed your comments raised in a previous round of review and you feel that this manuscript is now acceptable for publication, you may indicate that here to bypass the “Comments to the Author” section, enter your conflict of interest statement in the “Confidential to Editor” section, and submit your "Accept" recommendation.

Reviewer #3: (No Response)

Reviewer #4: (No Response)

2. Is the manuscript technically sound, and do the data support the conclusions?

Reviewer #3: Partly

Reviewer #4: Partly

3. Has the statistical analysis been performed appropriately and rigorously? 

Reviewer #3: Yes

Reviewer #4: No

4. Have the authors made all data underlying the findings in their manuscript fully available?

Reviewer #3: Yes

Reviewer #4: Yes

5. Is the manuscript presented in an intelligible fashion and written in standard English?

Reviewer #3: Yes

Reviewer #4: Yes

6. Review Comments to the Author

Reviewer #3: Major revision:

1. The authors should add other pre-existed medical co-morbidities such as COPD, hepatic and renal diseases, etc. into the propensity score-matched pair method. For pre-existed medical co-morbidities, the authors can identify based on ICD codes from the NHIS database.

2. P6. The authors mentioned the study follow-up period (2010/11–2015/16).

P5. They also mentioned that 2010–2011 was used as the baseline years, while the follow-up period was until 2015–2016.

P3. This study aimed to examine the impact of the CRMHDP on

healthcare use and the health outcomes of patients using a population-based propensity score-matched retrospective cohort study over “eight” years.

These time periods are not consistent. Please specify exactly the start year and the end year of the follow-up. Most importantly, when is the implementation period for CRMHDP? To assess the effect of CRMHDP, the suitable time period should be before and after the implementation period for CRMHDP.

3. For health outcomes, it would be valuable to include data on the outcomes including admission, emergency department visits, or hospital stays during the follow-up period when comparing these two groups.

4. The authors can perform a sub-group analysis according to Hypertension and diabetes. Since the CRMHDP covers HTN and DM, it is essential to examine the effect separately.

5. The format of tables and figures needs to be revised as journal requested.

Reviewer #4: The authors present a manuscript investigating the effectiveness of chronic disease management for elderly patients with hypertension and/or Type 2 Diabetes mellitus. Using a population-based propensity score-matched retrospective cohort study, health outcomes and utilisation of healthcare in four cities in South Korea were investigated with a focus of estimation the effect of motivated physicians, patients and community support on these outcomes.

The manuscript has undergone a first revision.

The work per se is interesting and important but the statistical analysis and description is lacking information.

1.) First of all there should be some sample size considerations justifying why they included patients from 16 months‘ project (CRMHDP) time.

2.) The authors state that they have excluded patients with complications during the baseline year – what about death?

3.) In the section „study design and sample“ they state that the baseline period is September 1, 2010 to December 31, 2011. This is incorrectly stated in Fig. 1 – 2010 to 2010 and it is also misleading in the text where they write in the first sentence in the section „study design and sample“ 2010/2011 are used as baseline years. Why did not they start with the baseline period on January 1, 2010?

4.) The eight year coverage is also not clearly explained. If a patient is included in Sept 1, 2010 his/her observation period would be Sept 1, 2008 to Sept 1, 2015 and if a patient is included at the end of 2011 his/her observation period would be end of 2009 to end of 2016 – for me this are seven instead of 8 years (see also statistical analysis - during a period of two years prior to and five years after initial registration).

5.) As for propensity score matching the compliance of antihypertensive and antidiabetic drugs type DM during the two years prior to baseline years is included. How was this considered regarding inclusion/exclusion of patients. What is known about the duration of the diseases – how is this incorporated?

6.) They describe in a section the propensity score calculation and matching and then they have a section on independent variables – redundant?

7.) To show the quality of matching it would be good not only to have supplementary tables but to include a figure as e.g. in Riedl et al. Plos One 2016.

8.) How did the authors handle matching of patients in their statistical analysis – this information has to be included in the statistical analysis section.

9.) PS matching should result in two similar groups – therefore I would also like to see non-adjusted results beside the adjusted results.

10.) What about missing data and how were they handled?

11.) In the result section the interpretation has to be checked – increased, decreased seems to be confused sometimes and also the time period is not clear - one year before and after participation (preintervention period – postintervention period) – see table 1 for instance.

12.) More relevant literature from European countries should be included!

7. PLOS authors have the option to publish the peer review history of their article (what does this mean?). If published, this will include your full peer review and any attached files.

Reviewer #3: No

Reviewer #4: No

---

## [Author Response · Author response to Decision Letter 1]

28 Sep 2023

Reviewer #3 

Comments:

Q1. The authors should add other pre-existed medical co-morbidities such as COPD, hepatic and renal diseases, etc. into the propensity score-matched pair method. For pre-existed medical co-morbidities, the authors can identify based on ICD codes from the NHIS database.

Answer) As you mentioned, we couldn’t consider pre-existed co-morbidities besides hypertension and type 2 DM like COPD, hepatic and renal diseases, etc. However, patients diagnosed with catastrophic disease such as cancer, stroke, and myocardial infarction, rare and intractable disease were excluded to avoid confounding bias due to the differences in the presence of catastrophic disease between the intervention and control group. This would be limitation of our study. We also stated that on the discussion section. Modified text was marked by blue collar 

“… Moreover, we could not consider pre-existed co-morbidities besides hypertension and type 2 DM like COPD, hepatic and renal diseases, etc. However, patients diagnosed with catastrophic disease such as cancer, stroke, and myocardial infarction, rare and intractable disease were excluded to avoid confounding bias due to the differences in the presence of catastrophic disease between intervention and control group. ….” (discuss section)

Q2. 2. P6. The authors mentioned the study follow-up period (2010/11–2015/16).

P5. They also mentioned that 2010–2011 was used as the baseline years, while the follow-up period was until 2015–2016. P3. This study aimed to examine the impact of the CRMHDP on

healthcare use and the health outcomes of patients using a population-based propensity score-matched retrospective cohort study over “eight” years. These time periods are not consistent. Please specify exactly the start year and the end year of the follow-up. Most importantly, when is the implementation period for CRMHDP? To assess the effect of CRMHDP, the suitable time period should be before and after the implementation period for CRMHDP.

Answer) We made a mistake as writing the title words of this paper. As you mentioned, we replaced ‘over eight years’ with during five-year follow-up period in the title of the manuscript. This study used the past medication rates in two-year pre-intervention period as an confounding factor examing the effect of CRMHDP on the process and health outcome of health care service, which means that this study considers before and after the implementation period for CRMHDP. Modified text was marked by blue collar 

“The clinical variables were the presence of the coexistence of hypertension or type 2 DM, the past medication adherence prior to participation in the study, and the type of speciality by the physician treating the patients in the study. The past medication adherence rates during the two-year per-intervention period, were measured by the PDC. (Method section)” 

3. For health outcomes, it would be valuable to include data on the outcomes including admission, emergency department visits, or hospital stays during the follow-up period when comparing these two groups.

Answer) In this study, we used total mortality and hospital admission due to major complications in hypertension or type 2 DM as outcome variables. However, emergency department visits and hospital stays were not employed. It was included as an limitation of this study. 

“… Lastly, this study did not included emergency room visits and hospital stays as another health outcome ..” (discussion section)

4. The authors can perform a sub-group analysis according to Hypertension and diabetes. Since the CRMHDP covers HTN and DM, it is essential to examine the effect separately.

Answer) We did not examine the effect of the CRMHDP on hypertension and type 2 DM separately due to the following reasons. According to the existing literature , , various type of incentives including financial support to patient and physician can lead to improve the health outcome (BP or blood glucose control) or medication adherence of hypertension and type 2 DM. This study assumed the CRMHDP would have worked type 2 DM as well as hypertension simultaneously. Moreover, type 2 DM and essential hypertension are common conditions that are frequently present together. Patients with type 2 DM combining hypertension accounted for about 25% of the entire subject of this study. However, since our assumption of the effect of the CRMHDP on them may be wrong, the results of this study should be interpreted very cautiously. We added this text to the discussion section as a limitation of this study. 

5. The format of tables and figures needs to be revised as journal requested.

Answer) Yes, we revised those tables and figures as journal requested.

Reviewer #4: The authors present a manuscript investigating the effectiveness of chronic disease management for elderly patients with hypertension and/or Type 2 Diabetes mellitus. Using a population-based propensity score-matched retrospective cohort study, health outcomes and utilisation of healthcare in four cities in South Korea were investigated with a focus of estimation the effect of motivated physicians, patients and community support on these outcomes.

The manuscript has undergone a first revision.

The work per se is interesting and important but the statistical analysis and description is lacking information.

1.) First of all there should be some sample size considerations justifying why they included patients from 16 months‘ project (CRMHDP) time.

Answer) We made change of the relevant text in stud design and sample of method section as follows.

“ the CRMHDP group (n = 48,325), designated as the intervention group, was composed of patients who enrolled in the project via participating local clinics (n = 409) which accounted for 84.2% of all local clinics (n = 486), in the cities, during the baseline period (September 1, 2010, to December 31, 2011). The local clinics not engaged in the project had few patients with hypertension and type 2 DM because their clinical practitioners were specialist such as opthamologist, pediatric physician, otolaryngologist and opthalmologis. Therefore, the registered patients can represent all elderly hypertensive and diabetic patient in the cities.”

2.) The authors state that they have excluded patients with complications during the baseline year – what about death?

Answer ) We excluded died patient from this study. It was added to exclusion criteria in the method section 

3.) In the section „study design and sample“ they state that the baseline period is September 1, 2010 to December 31, 2011. This is incorrectly stated in Fig. 1 – 2010 to 2010 and it is also misleading in the text where they write in the first sentence in the section „study design and sample“ 2010/2011 are used as baseline years. Why did not they start with the baseline period on January 1, 2010?

Answer) We corrected the misspelling in Figure 1(2010). Registration to the project was recommended a elderly patient with hypertension and type 2 DM aged 65 and over by a physician at local clinic. It took one and half year for 48,325 subjects of the project to were registered in local clinics. 

4.) The eight year coverage is also not clearly explained. If a patient is included in Sept 1, 2010 his/her observation period would be Sept 1, 2008 to Sept 1, 2015 and if a patient is included at the end of 2011 his/her observation period would be end of 2009 to end of 2016 – for me this are seven instead of 8 years (see also statistical analysis - during a period of two years prior to and five years after initial registration).

Answer) We made a mistake as writing the title words of this paper. As you mentioned, we replaced ‘over eight years’ with during five-year follow-up period in the title of the manuscript. 

5.) As for propensity score matching the compliance of antihypertensive and antidiabetic drugs type DM during the two years prior to baseline years is included. How was this considered regarding inclusion/exclusion of patients. What is known about the duration of the diseases – how is this incorporated?

Answer) This study used the past medication rates in two-year pre-intervention period as an confounding factor examing the effect of CRMHDP on the process and health outcome of health care service, which means that this study considers before and after the implementation period for CRMHDP. However, we did not consider the duration of the disease because it could not be identified in the KNHIS (Korean National Health Insurance) data. We included it as an study limitation of the discussion section. Modified text was marked by blue collar. 

6.) They describe in a section the propensity score calculation and matching and then they have a section on independent variables – redundant?

Answer) The section the propensity score calculation addressed independent variables as matching one whereas the section independent variable explained how to measure them. We think that to keep it would be helpful for journal readers 

7.) To show the quality of matching it would be good not only to have supplementary tables but to include a figure as e.g. in Riedl et al. Plos One 2016.

Answer) As one of the methods of evaluating the quality of matching, our study presented the standardized difference in measures (Table S1, Table S2), confirming that the difference between the intervention and control group was less than 10%. As a result, we have suggested through the result in the tables that the matching has been properly balanced. We also show the additional box plot to show the quality of matching properly (see figure 4). 

8.) How did the authors handle matching of patients in their statistical analysis – this information has to be included in the statistical analysis section.

Answer) The description of the matching is explained in detail in the ‘Propensity score calculation and matching’ section (manuscript page 6). 

9.) PS matching should result in two similar groups – therefore I would also like to see non-adjusted results beside the adjusted results.

Answer) Table S1, Table S2 contain statistical result and statistical test for variables before and after matching (see supplement Table S1 and Table S2). 

10.) What about missing data and how were they handled?

Answer) If there is missing in the variable or there is no control patient within the proper range of the propensity score, these data are eliminated all. Those data are not included in the neither intervention nor control group in this study. Therefore, there is no missing in the matching variables of the data used in the analysis. 

11.) In the result section the interpretation has to be checked – increased, decreased seems to be confused sometimes and also the time period is not clear - one year before and after participation (preintervention period – postintervention period) – see table 1 for instance.

Answer) We modified the sentence as written below. Modified text was marked by blue collar in the manuscript. 

This implied that the intervention increased the annual clinic visits per person one year before and after the intervention.

-> “This implied that the intervention contributed to escalate the annual clinic visits per person one year before and a year after the intervention.” (Results section, page 8) 

This meant that the intervention led to an increase in the annual days covered for antihypertensive, antidiabetic drug and insulin injections.

-> “This meant that the intervention led to strengthen compliance in the annual days covered for antihypertensive, antidiabetic drug and insulin injections.” (Results section, page 8) 

12.) More relevant literature from European countries should be included!

Answer) We included some more relevant European literature in the text. Modified text was marked by blue collar in reference list. (Reference section)

---

## [Decision Letter · Decision Letter 2]

2 Oct 2023

PONE-D-23-13912R2Effect of motivated physicians and elderly patients with hypertension or type 2 diabetes mellitus in prepared communities on health behaviours and outcomes: a population-based propensity score-matched retrospective cohort study over eight yearsPLOS ONE

Dear Dr. Lee,

Thank you for submitting your manuscript to PLOS ONE. After careful consideration, we feel that it has merit but does not fully meet PLOS ONE’s publication criteria as it currently stands. Therefore, we invite you to submit a revised version of the manuscript that addresses the points raised during the review process.

Please attempt to address the comments of the two reviewers. ==============================

We look forward to receiving your revised manuscript.

Kind regards,

Yee Gary Ang, MBBS MPH

Academic Editor

PLOS ONE

Journal Requirements:

Reviewers' comments:

Reviewer's Responses to Questions

**Comments to the Author**

1. If the authors have adequately addressed your comments raised in a previous round of review and you feel that this manuscript is now acceptable for publication, you may indicate that here to bypass the “Comments to the Author” section, enter your conflict of interest statement in the “Confidential to Editor” section, and submit your "Accept" recommendation.

Reviewer #3: (No Response)

Reviewer #4: (No Response)

2. Is the manuscript technically sound, and do the data support the conclusions?

Reviewer #3: Partly

Reviewer #4: Yes

3. Has the statistical analysis been performed appropriately and rigorously? 

Reviewer #3: No

Reviewer #4: No

4. Have the authors made all data underlying the findings in their manuscript fully available?

Reviewer #3: Yes

Reviewer #4: Yes

5. Is the manuscript presented in an intelligible fashion and written in standard English?

Reviewer #3: Yes

Reviewer #4: Yes

6. Review Comments to the Author

Reviewer #3: For using administrative database, it is weird that the authors have difficulties to incorporate other pre-existing comorbidities. Except this, however, the authors have not adequately addressed my comments regarding a sub-group analysis according to Hypertension and diabetes. The authors mentioned only 1/4 of individuals with both HTN and DM. Hence, it is adequate to perform the sub-group analysis to examine the effect in more details.

Reviewer #4: The answers to most of my questions were appropriate.

The answer to the question how the authors handled matching of patients in their statistical analysis was not adequate. The matching and the ratio of 1:2 has to be considered in the statistical analysis section (use the appropriate variance estimator) and not in the matching process (Propensity score calculation and matching’ section).

Concerning missing data: it should be made clear what they have done- that means they should include their answer in the text. "" If there is missing in the variable or there is no control patient within the proper range of the propensity score, these data are eliminated all. Those data are not included in the neither intervention nor control group in this study. Therefore, there is no missing in the matching variables of the data used in the analysis."

The authors have to check the references - the journal names are not included?

7. PLOS authors have the option to publish the peer review history of their article (what does this mean?). If published, this will include your full peer review and any attached files.

Reviewer #3: No

Reviewer #4: No

---

## [Author Response · Author response to Decision Letter 2]

14 Nov 2023

As requested by National Health Insurance Service (NHIS), the data provider, the research number below is listed on the Manuscript. 

Acknowledgement: This study used NHIS-NSC data (NHIS-2020-1-178) from the National Health Insurance Service (NHIS). 

Reviewer #3 

Comments:

Q1. For using administrative database, it is weird that the authors have difficulties to incorporate other pre-existing comorbidities. Except this, however, the authors have not adequately addressed my comments regarding a sub-group analysis according to Hypertension and diabetes. The authors mentioned only 1/4 of individuals with both HTN and DM. Hence, it is adequate to perform the sub-group analysis to examine the effect in more details.

Answer) We can understand your concerns regarding to not making a sub-group analysis according to Hypertension and Diabetes. However, according to the huge existing literature (36-38) related to health behavioural economics, financial incentives to chronic patient and physician for the change of patient behavior has been very effective tool at changing behaviors in chronic condition related to medication adherence. Hence, this study assumed the CRMHDP would have positive impact on patient behavior including medication adherence among both hypertensive patients and diabetic patients. However, since our assumption of the CRMHDP’s impact on them may be wrong, a sub-group analysis should be made in the further. We changed the text of discussion section as mentioned above. 

Added the citation also in the reference list

36) Hohmann NS, Hastings TJ, Jeminiwa RN, Qian J, Hansen RA, Ngorsuraches S, Garza KB. Patient preferences for medication adherence financial incentive structures: A discrete choice experiment. Res Social Adm Pharm. 2021 Oct;17(10):1800-1809. doi: 10.1016/j.sapharm.2021.01.018. Epub 2021 Feb 5. PMID: 33608244; PMCID: PMC9070307.

37) Adams AS, Madden JM, Zhang F, Soumerai SB, Gilden D, Griggs J, Trinacty CM, Bishop C, Ross-Degnan D. Changes in use of lipid-lowering medications among black and white dual enrollees with diabetes transitioning from Medicaid to Medicare Part D drug coverage. Med Care. 2014 Aug;52(8):695-703. doi:

38) Kong E, Beshears J, Laibson D, Madrian B, Volpp K, Loewenstein G, Kolstad J, Choi JJ. Do physician incentives increase patient medication adherence? Health Serv Res. 2020 Aug;55(4):503-511. doi: 10.1111/1475-6773.13322. PMID: 32700389; PMCID: PMC7376007.

Reviewer #4: The answers to most of my questions were appropriate. 

Q1. The answer to the question how the authors handled matching of patients in their statistical analysis was not adequate. The matching and the ratio of 1:2 has to be considered in the statistical analysis section (use the appropriate variance estimator) and not in the matching process (Propensity score calculation and matching’ section).

Answer) We included the text about the matching and the ratio of 1:2 in the statistical analysis section, as you mentioned above (page 6). 

The introduction is as follows, 

“ The analysis was based on propensity score matching(PSM) which was first proposed by Rosenbaum and Rubin in the 1980 (17) ~”

Added the citation also in the reference list

(17) Rosenbaum PR, Rubin DBJB. The central role of the propensity score in observational studies for causal effects. 1983;70(1):41-55.

Q2. Concerning missing data: it should be made clear what they have done- that means they should include their answer in the text. "" If there is missing in the variable or there is no control patient within the proper range of the propensity score, these data are eliminated all. Those data are not included in the neither intervention nor control group in this study. Therefore, there is no missing in the matching variables of the data used in the analysis."

Answer ) We included our answer about missing data evidently in the statistical analysis section. (page 7, 9-11 lines) 

"If there is missing in the variable or there is no control patient within the proper range of the propensity score, these data are eliminated all. Those data are not included in the neither intervention nor control group in this study. Therefore, there is no missing in the matching variables of the data used in the analysis. "

Q3. The authors have to check the references - the journal names are not included?

Answer ) The references are organized in Vancouver style using End Note program according to the journal’s reference style requirement. The additional literature has also been rearranged in the same.

---

## [Decision Letter · Decision Letter 3]

20 Dec 2023

Effect of motivated physicians and elderly patients with hypertension or type 2 diabetes mellitus in prepared communities on health behaviours and outcomes: a population-based PS matched retrospective cohort study during five-year follow-up period

PONE-D-23-13912R3

Dear Dr. Lee,

We’re pleased to inform you that your manuscript has been judged scientifically suitable for publication and will be formally accepted for publication once it meets all outstanding technical requirements.

There was a difference in opinions by the 2 reviewers and a third reviewer was invited to review the manuscript.

In the end, I am satisfied the authors have addressed all necessary revisions required prior to publication. 

Kind regards,

Yee Gary Ang, MBBS MPH

Academic Editor

PLOS ONE

Reviewers' comments:

Reviewer's Responses to Questions

**Comments to the Author**

1. If the authors have adequately addressed your comments raised in a previous round of review and you feel that this manuscript is now acceptable for publication, you may indicate that here to bypass the “Comments to the Author” section, enter your conflict of interest statement in the “Confidential to Editor” section, and submit your "Accept" recommendation.

Reviewer #3: (No Response)

Reviewer #4: All comments have been addressed

Reviewer #5: All comments have been addressed

2. Is the manuscript technically sound, and do the data support the conclusions?

Reviewer #3: No

Reviewer #4: (No Response)

Reviewer #5: Yes

3. Has the statistical analysis been performed appropriately and rigorously? 

Reviewer #3: No

Reviewer #4: (No Response)

Reviewer #5: I Don't Know

4. Have the authors made all data underlying the findings in their manuscript fully available?

Reviewer #3: No

Reviewer #4: (No Response)

Reviewer #5: Yes

5. Is the manuscript presented in an intelligible fashion and written in standard English?

Reviewer #3: Yes

Reviewer #4: (No Response)

Reviewer #5: Yes

6. Review Comments to the Author

Reviewer #3: After two-round of revisions, the authors still have not replied to my suggestions adequately. The first issue is to deal with other confounders, such as comorbidities, in their analysis. The second part is related to sub-group analysis which could determine the robustness of their findings.

In its current state, I do not recommend accepting this paper.

Reviewer #4: (No Response)

Reviewer #5: well done; all revisions completed as required

7. PLOS authors have the option to publish the peer review history of their article (what does this mean?). If published, this will include your full peer review and any attached files.

Reviewer #3: No

Reviewer #4: No

Reviewer #5: No

---

## [Editor Report · Acceptance letter]

27 Jan 2024

PONE-D-23-13912R3 

PLOS ONE

Dear Dr. Lee, 

I'm pleased to inform you that your manuscript has been deemed suitable for publication in PLOS ONE. Congratulations! Your manuscript is now being handed over to our production team.

Kind regards, 

on behalf of

Dr. Yee Gary Ang 

Academic Editor

PLOS ONE